# Oilfield Reservoir Parameter Inversion Based on 2D Ground Deformation Measurements Acquired by a Time-Series MSBAS-InSAR Method

Anmengyun Liu [1], Rui Zhang [1,2,*], Yunjie Yang [1], Tianyu Wang [1], Ting Wang [1], Age Shama [1], Runqing Zhan [1] and Xin Bao [1]

1   Faculty of Geosciences and Environmental Engineering, Southwest Jiaotong University, Chengdu 611756, China; liuann@my.swjtu.edu.cn (A.L.); 2395565241@my.swjtu.edu.cn (Y.Y.); tianyuwang@my.swjtu.edu.cn (T.W.); wangting@my.swjtu.edu.cn (T.W.); shamaage@my.swjtu.edu.cn (A.S.); zhanrunqing@my.swjtu.edu.cn (R.Z.); baoxin@my.swjtu.edu.cn (X.B.)
2   State-Province Joint Engineering Laboratory of Spatial Information Technology for High-Speed Railway Safety, Southwest Jiaotong University, Chengdu 611756, China
*   Correspondence: zhangrui@swjtu.edu.cn

**Abstract:** Time-series ground deformation monitoring and reservoir parameter inversion are crucial for the dynamic assessment of oilfield resources and sustainable exploitation in oilfields. As some of the regions with the richest oil reserves in China, the oilfield areas in the western Qaidam Basin were selected as a typical study area. Firstly, we used SAR images collected by the Sentinel-1A satellite from January 2021 to December 2022 and applied the multidimensional small baseline subset (MSBAS) method to obtain vertical and east–west deformation measurements. On this basis, a nonlinear Bayesian inversion method was applied to model the shallow reservoir in a series of complex deformation areas, based on a single-source model and a multi-source model, respectively. As a result, the ground deformation monitoring results obtained by long time-series InSAR clearly reflect the uneven ground deformation caused by the oil extraction and water injection operation processes. There was slight subsidence in the Huatugou oilfield, while significant uplift deformation occurred in the Ganchaigou oilfield and the Youshashan oilfield, with a maximum uplift rate of 48 mm/year. Further analysis indicated that the introduction of the 2D deformation field helps to improve the robustness of oilfield reservoir parameter inversion. Moreover, the dual-source model is more suitable than the single-source model for inverting reservoir parameters of complex deformation. This study not only fills the gap of InSAR deformation monitoring for the oilfields in the western Qaidam Basin but also provides a theoretical reference for the model and method selection of reservoir parameter inversion in other oilfields.

**Keywords:** oilfield deformation; MSBAS-InSAR; Qaidam Basin; geophysical models



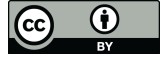

## 1. Introduction

As oil is the world's primary energy source, its extraction and storage are of great significance to national economic development. However, it also brings environmental problems that cannot be ignored [1–3]. Oil extraction can cause not only subsidence but also uplift. This is because fluid injection into the reservoir is often carried out to enhance oil recovery, resulting in excessive underground reservoir pressure, which manifests as a ground uplift [4,5]. Whether subsidence or uplift, severe ground deformation not only damages oil well production facilities but also affects the safety of surrounding infrastructure such as buildings, railways, and roads. Therefore, ground deformation monitoring and reservoir parameter inversion in oil extraction areas are essential for ensuring the safety of oilfields. The Qaidam Basin, often referred to as China's "treasure trove", is not only abundant in various mineral resources but also hosts important oil and gas accumulation

zones in its western region [6,7]. By the year 2020, the Qaidam Basin boasted 32 identified oil and gas fields, with combined proven reserves exceeding 6.4 billion tons [8,9]. Previous studies in this area mostly described the reservoir characteristics of oilfields from the perspectives of geology, sedimentary petrology, and geochemistry [10,11]; there has not been any research on utilizing interferometric synthetic-aperture radar (InSAR) technology for deformation characteristic monitoring and reservoir parameter inversion. Therefore, using InSAR technology to monitor the deformation of the oilfields in the western basin not only provides guidance for sustainable oil extraction but also offers a new approach to understanding the reservoir characteristics in this region.

Conventional monitoring techniques require significant time and effort. However, InSAR technology, due to its benefits of extensive coverage, all-weather capabilities, precision, and efficiency, has found broad application in the monitoring of urban ground deformation [12–14], as well as in the monitoring and early warning of geological disasters such as earthquakes, landslides, and others [15–17]. Currently, the most commonly employed InSAR techniques include permanent scatterer InSAR (PS-InSAR) [18], small baseline subset InSAR (SBAS-InSAR) [19], and distributed scatterer InSAR (DS-InSAR) [20–22], which can not only effectively overcome the shortcomings of phase decoherence, terrain residual, and atmospheric delay [23], but also quickly obtain the deformation time series in the line-of-sight (LOS) direction [24]. Meanwhile, some scholars have also applied InSAR technology to monitor ground deformation caused by underground reservoir changes in oilfields, geothermal fields, and mines, thus exposing the intimate connection between ground deformation characteristics and the state of underground reservoirs [25–27]. Sun et al. [28] used multitrack PS-InSAR technology to quantitatively analyze three types of ground subsidence, including oilfields, in the lower Liaohe Plain of China. Juncu et al. [29] extracted deformations in a geothermal field using SBAS-InSAR technology and found that the primary factor contributing to the subsidence was the observed pressure drawdown. Chen et al. [30] used differential InSAR (D-InSAR) and SBAS technology to acquire cumulative deformation data within a mining area in China and compared them with the progress of mining operations. However, one-dimensional deformation measurement can not only fail to provide more comprehensive and intuitive information for detecting the ground deformation characteristics of oilfields but also struggle to reflect the real ground deformation conditions, often leading to deformation interpretation deviations, which is called the LOS fuzzy problem.

In addition, for oilfields, changes in reservoir pore pressure caused by fluid extraction and injection, as well as the geometric shapes, positions, volumes, and other physical parameters of the reservoir, greatly influence the ground deformation characteristics. Therefore, using InSAR deformation results to invert reservoir parameters in oilfields can provide a quick and timely understanding of the reservoir's state, which is of great significance for oilfield stability evaluation. Klemm et al. [31] monitored an oilfield in the Middle East with PS-InSAR and inverted ground deformations using a geomechanical model. They found that monitoring ground deformations and applying geomechanical inversion can yield valuable insights into the dynamic characteristics of reservoirs. Yang et al. [32] obtained subsidence information of a hydrocarbon reservoir in West Texas, USA using the SBAS method and simulated ground subsidence based on reservoir parameters. By comparing the InSAR-observed deformations with the modeled deformations, they found a high degree of agreement between the two. Ji et al. [5] conducted subsidence monitoring in the Karamay oilfield in Xinjiang using the SBAS method, and then they inverted its reservoir's geometric parameters based on the Okada model. Nevertheless, there is still limited research on applying InSAR deformation results to reservoir parameter inversion in oilfields. The inversion theory is relatively lacking, and most scholars use a single model for the inversion under the assumption that there exists only one source of deformation in the subsurface. There is a lack of comparative studies on joint inversion of multi-source models for complex deformations. Moreover, some scholars have jointly inverted the seismic source parameters with the LOS and azimuthal deformation fields of ascending

and descending images to improve the inversion efficiency and accuracy by increasing the constraints of nonlinear inversion [33,34], but there have been no similar attempts to date in the field of subsurface reservoir parameter inversion, and they all directly utilized the one-dimensional deformation in the inversion.

Based on the 113 ascending and descending Sentinel-1A SAR images from January 2021 to December 2022, multidimensional small baseline subset (MSBAS) InSAR technology was employed in this study to determine the vertical and east–west ground deformation of the oilfields in the western Qaidam Basin and then analyze its temporal and spatial characteristics and causes. Afterwards, we combined the obtained two-dimensional deformation measurements for constraints and used a single-source model and a dual-source model to invert the reservoir parameters of the two complex deformation areas in the Youshashan oilfield within the study area, and then we conducted comparative analysis and evaluation of the inversion results. The related results can provide references for the safety maintenance of oilfields in the western Qaidam Basin and the selection of reservoir parameter inversion models and methods in other areas.

## 2. Study Area and Datasets

### 2.1. Background of the Study Area

Located in the northern part of the Qinghai–Tibet Plateau, the Qaidam Basin is one of four major basins in China. It is surrounded by the Kunlun Mountains, Qilian Mountains, Altyn Mountains, and other mountains, covering an expanse of about 240,000 square kilometers. Within the basin, there are not only extensive salt lakes and marshes but also abundant reserves of oil, coal, and various metallic minerals.

The study area is located in the western part of the Qaidam Basin in China; the terrain is high in the east and low in the west, with an average altitude of about 3000 m (Figure 1). The climate there is dry and cold, with less rain and more wind, long winters and short summers, four unclear seasons, and large daily temperature differences, representing a typical temperate continental climate. In this region, the Paleocene and Neocene systems are mainly developed, and the Quaternary is uplifted and thinly deposited. The reservoirs can be broadly categorized into two types: porous clastic rocks and fractured, pore-vug diamictites. In the vertical direction, these two reservoir types are alternately stacked, forming a reservoir distribution characteristic that develops continuously in time and alternately overlaps in space. The unique geological and geographic conditions have fostered abundant oil resources. Following the commencement of development in the Youshashan (YSH) oilfield in 1957, the Huatugou (HTG) oilfield, the Youyuangou (YYG) oilfield, the Shizigou (SZG) oilfield, and the Ganchaigou (GCG) oilfield were subsequently developed within the study area. At the same time, the fluid injection technique was employed to increase oilfield production. By 2014, the total oil production in this region had exceeded 6.5 million tons [35].

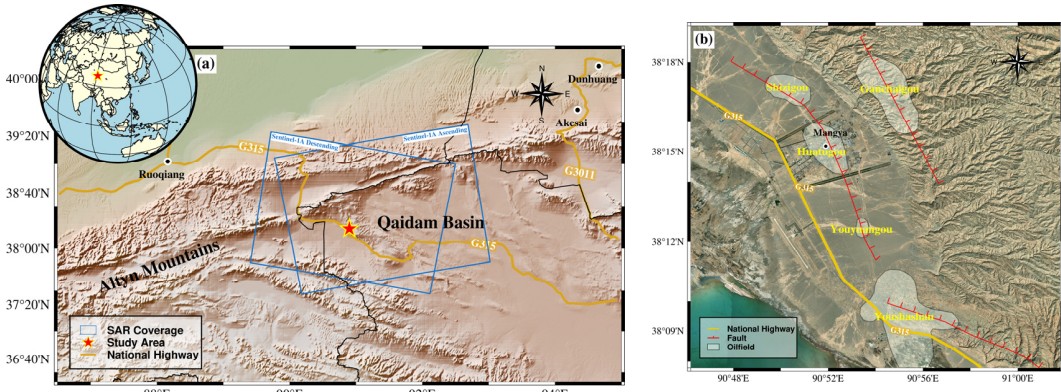

**Figure 1.** (**a**) Overview of the western part of the Qaidam Basin overlaid on the topography. (**b**) Detailed satellite image of the study area. The oilfields' boundaries are depicted by the white polygons.

## 2.2. Data and Processing

The dataset used in this study consists of 58 ascending and 56 descending Sentinel-1A SAR images covering the period from January 2021 to December 2022 (Table 1). Sentinel-1A is a satellite equipped with C-band synthetic-aperture radar, with an orbital height of 690 km and a revisit period of 12 days. Strict orbital control technology is adopted to ensure that the space baseline is small enough to improve the SAR image interference effect. GAMMA software [36] was used for D-InSAR processing in this study. Terrain errors were removed by the digital elevation model (DEM) of the Shuttle Radar Topographic Mapping Mission (SRTM) with 30 m resolution. At the same time, to remove the atmospheric delay, the Generic Atmospheric Correction Online Service (GACOS) data were also introduced. Finally, to improve the quality of the InSAR results, the interference pairs with spatiotemporal baselines less than 200 m and 49 days (Figure 2) were selected to perform the phase unwrapping via the minimum-cost flow (MCF) method.

**Table 1.** Main parameters of the SAR data.

| Sensor | Wavelength | Azimuth/Range Pixel Spacing | Orbit Direction | Path | Temporal Coverage |
|---|---|---|---|---|---|
| Sentinel-1A | 5.6 cm | 13.99 m/2.33 m | Ascending | 143 | 11 January 2021–20 December 2022 |
| | | | Descending | 48 | 5 January 2021–26 December 2022 |

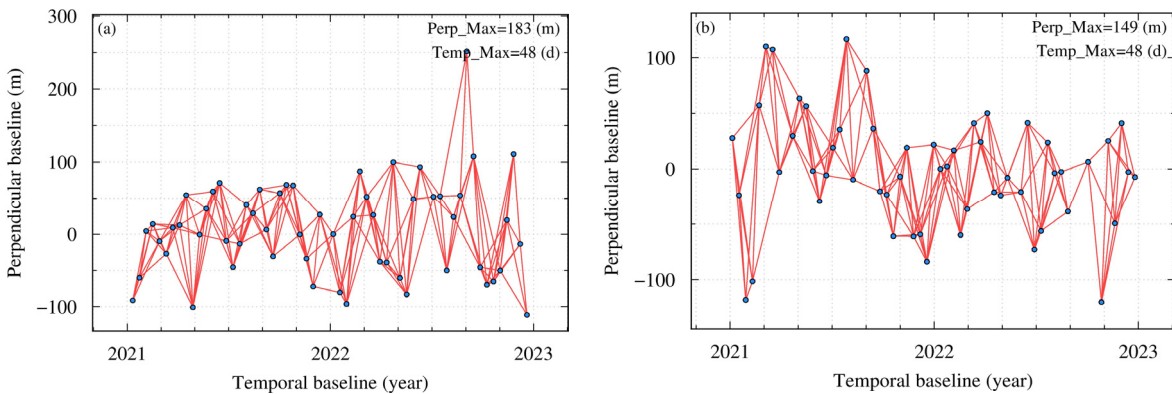

**Figure 2.** Spatiotemporal baseline of the (**a**) ascending and (**b**) descending interfering pairs.

## 3. Methodology

### 3.1. MSBAS-InSAR Method

MSBAS technology is an improved version of SBAS technology. The result generated by SBAS is the time series of LOS displacement for each pixel in an SAR image. In the case of a single acquisition geometry without supplementary data, it is not possible to fully decompose the LOS solution obtained from SBAS into its three components of ground displacement (vertical, east–west, and north–south directions) [37]. However, when the InSAR data come from multiple acquisition geometries, the SBAS method can be adjusted to generate an approximate solution that includes time series of multiple components [38]. Therefore, by simultaneously processing multiple D-InSAR orbit data, MSBAS can obtain the vertical and horizontal east–west deformation time series of the overlapping area [39]. The main steps of the MSBAS method are as follows: Firstly, D-InSAR processing is performed on one or more sets of ascending and descending orbit data, followed by resampling the unwrapped and geocoded differential interferograms to a consistent area and grid size. Secondly, a temporal matrix is created and decoherent pixels are removed. Finally, each pixel is solved by singular value decomposition (SVD) [40] to acquire the two-dimensional deformation rate component, and then the deformation time series can be reconstructed by numerically integrating the deformation rate. The use of Tikhonov

regularization during the matrix-solving process helps solve ill-posed problems in the observation equation. The inversion matrix is

$$\begin{pmatrix} -\cos\alpha\sin\psi\Delta t & \cos\psi\Delta t \\ \lambda I \end{pmatrix}\begin{pmatrix} V_E \\ V_U \end{pmatrix} = \begin{pmatrix} \hat{\phi} \\ 0 \end{pmatrix} \tag{1}$$

where $\psi$, $\alpha$ represent the incident and azimuth angle, respectively, $\Delta t$ represents the time interval between adjacent SAR images, $\lambda$ is a regularization parameter, $I$ represents the identity matrix, $V_E$ and $V_U$ are the east–west and vertical deformation rate, respectively, and $\hat{\phi}$ represents the obtained differential interferometry measurements. If we denote the coefficient matrix by $C$

$$C = \begin{pmatrix} -\cos\alpha\sin\psi\Delta t & \cos\psi\Delta t \\ \lambda I \end{pmatrix} \tag{2}$$

then we can obtain the east–west and vertical deformation rates using the following formula:

$$\begin{pmatrix} V_E \\ V_U \end{pmatrix} = (C^T C)^{-1} C^T \begin{pmatrix} \hat{\phi} \\ 0 \end{pmatrix} \tag{3}$$

### *3.2. The Source Model and Inversion Method*

In 1977, Matsu'Ura [41] inverted the fault properties of the Tango earthquake based on geodetic data and clearly proposed the concept of "geodetic inversion" for the first time. Geodetic inversion mainly consists of three components: geodetic data, a geophysical inversion model, and an inversion algorithm. In this study, the deformation data obtained through InSAR technology were used as geodetic data, and appropriate inversion models and algorithms were employed to obtain the underground reservoir parameters of the deformation area in the oilfields. Next, the geophysical inversion models and inversion algorithms used in this study will be introduced.

### 3.2.1. Finite Prolate Spheroidal Model

In 1988, Yang et al. [42] derived and calculated the analytical formula for the arbitrary-oriented prolate spheroidal cavity model within a finite-dimensional elastic half-space. They proposed the finite prolate spheroidal model and found that this model has relatively more parameter degrees of freedom compared to the ellipsoidal point-source model. The finite prolate spheroidal model is composed of eight parameters, including the three-dimensional coordinates of the ellipsoid's center, the long axis, the short axis, and pressure variation, as well as the strike and dip of the long axis. A spatial rectangular coordinate system with O as the origin is established, as shown in Figure 3. M represents the coordinates $(x_0, y_0, -d)$ of the ellipsoid's source center, a and b are the major and minor semi-axes of the ellipsoid, respectively, $\varphi$ is the strike, and $\theta$ is the dip angle.

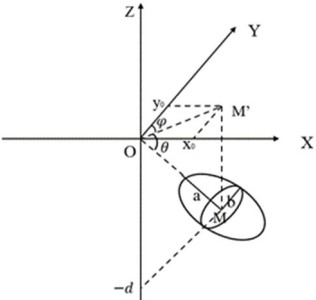

**Figure 3.** Schematic diagram of the finite prolate spheroidal model.

### 3.2.2. Dipping Dike with Uniform Opening Model

In 1985, Japanese scholar Yoshimitsu Okada [43] analyzed existing research results on ground deformation caused by elastic half-space faults and proposed a general expression

for ground deformation caused by fault dislocations of point sources and finite rectangular surface sources. In 1992, Okada [44] further improved the theory of this fault dislocation model. According to the elastic half-space isotropic dislocation theory, the displacement of a point on the ground due to the dislocation of a rectangular surface within an elastic medium is proportional to the amount of displacement of that surface. The proportionality coefficient is uniquely determined by the relative positions, the geometric dimensions, the inclination, the depth, and the elastic medium of the dislocation surface. If there are multiple dislocation surfaces underground, then the displacement of the point on the ground is the sum of the displacement vectors caused by the respective dislocations of the multiple dislocation surfaces.

This method establishes a spatial rectangular coordinate system with O as the origin, as shown in Figure 4. There are seven main parameters of the dipping dike with uniform opening model, including the length along the strike (L), the width along the inclination (W), the depth of the geometric center of the dislocation surface (d), the inclination angle ($\theta$), the strike ($\varphi$), and the projected coordinates ($x_0$, $y_0$) of the geometric center point M on the ground surface.

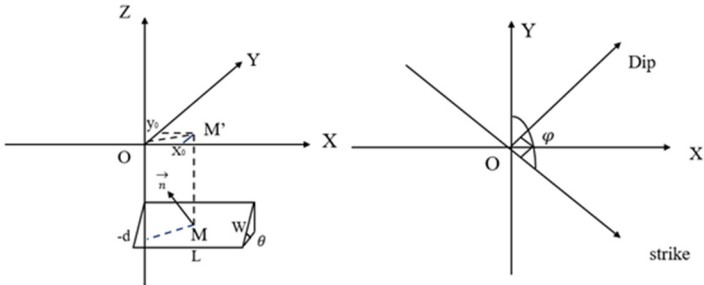

**Figure 4.** Schematic diagram of the Okada model.

3.2.3. The Nonlinear Bayesian Inversion Method

Geodetic inversion is a discipline that studies the evolutionary characteristics and laws of ground deformation based on geodetic observation data combined with prior information, and then infers the internal and physical parameters of the earth, thereby revealing the internal dynamics of the earth [45]. In the process of geodetic inversion, the functional relationship between observed data and model parameters is as follows:

$$d = G(m) + \varepsilon \tag{4}$$

where $d$ denotes the observed data, $m$ represents the model parameters, $G$ is the Green's function that connects the model parameters and ground deformation, and $\varepsilon$ is the model error.

Instead of the traditional deterministic inversion algorithm that determines the optimal model parameters by minimizing the error between observation data and model simulation data, the nonlinear Bayesian inversion algorithm was the stochastic inversion algorithm used in this paper [46]. By obtaining the posterior probability density function (PDF) of all model parameters, this algorithm can not only determine the optimal parameter values but also evaluate the uncertainty of the parameter values well, making it more practical than the traditional algorithms. In addition, this paper proposes utilizing the 2D deformation field measured by MSBAS to constrain and invert the vertical and east–west deformations of the oilfields, so as to obtain more reliable reservoir parameters of the oilfield. At the same time, quadtree sampling was applied to improve the inversion efficiency. The likelihood function $p(d \mid m)$ can be expressed as follows:

$$p(d|m) = (2\pi)^{-N/2} \left| \sum_d \right|^{-\frac{1}{2}} \times \exp\left[ -\frac{1}{2}(d - Gm)^T \sum_d^{-1} (d - Gm) \right] \tag{5}$$

where $d = [d_u, d_e]^T$ represents the vertical and east–west observation data, $m$ represents the model parameters, $N$ refers to the number of data points, $\Sigma_d$ refers to the variance–covariance matrix of the data, and $Gm = [G_u m, G_e m]^T$ represents the 2D simulated data calculated from the model parameters $m$ through the Green's function $G$.

After obtaining the prior information and likelihood function, the PDF $p(m|d)$ can be computed by the Bayesian formula. This refers to the probability of the current parameter effectively accounting for the observed data when considering the prior information.

In addition, the Markov chain Monte Carlo (MCMC) sampling method and the Metropolis–Hastings rule [47,48] were used to randomly modify the model parameters and calculate new likelihood function values. The newly calculated likelihood function values were compared with the previously computed values to decide whether the current model parameters should be approved or rejected. This process was iterated until the termination condition was met.

## 4. Results

### 4.1. Two-Dimensional (2D) Deformation Monitoring and Analysis

The MSBAS method was employed in this research to simultaneously process ascending and descending SAR images, leading to the generation of a vertical average annual deformation rate map and an east–west average annual deformation rate map of the study area, as depicted in Figure 5a and b, respectively. It can be observed that from January 2021 to December 2022, the GCG oilfield, HTG oilfield, and YSS oilfield in the study area exhibited varying degrees of ground deformation. Specifically, the GCG oilfield and YSS oilfield experienced notable ground uplift and east–west deformation, with the highest uplift rate reaching 48 mm/year. Only uneven ground subsidence occurred in the HTG oilfield, without significant horizontal displacement, and with a maximum subsidence rate of about 12 mm/year. Previous studies have indicated that the injection of fluid and gas during oil extraction can cause ground uplift [25]. The study area is characterized by multiple faults and complex structures, posing challenges in oil production. To overcome these challenges and achieve increased and sustained production, water injection methods have been widely adopted in the oilfields. Therefore, the ground uplift in the GCG oilfield and YSS oilfield may be associated with the oilfields' water injection activities. However, although the HTG oilfield employs the same methods to enhance production, its ground did not rise but subsided. This may have been due to the fact that the HTG oilfield is located in the urban area of Mangya, where residents predominantly depend on groundwater extraction for their daily water supply. The substantial groundwater and oil extraction, along with urban construction, has resulted in the compaction of the ground in this area. Considering the complexity of the causes of ground subsidence in the HTG oilfield, this study focuses only on discussing and analyzing the deformation characteristics of the uplift areas (Area A, Area B, and Area C) within the GCG oilfield and the YSS oilfield.

As shown in Figure 5, Area A exhibits a maximum uplift rate of 16 mm/year, along with a maximum east–west deformation rate of 8 mm/year. Most of Area B moves westward, and there is an eastward displacement in a small local area. The maximum uplift rate reaches 30 mm/year, and the maximum east–west deformation rate reaches 7 mm/year. The maximum uplift rate of Area C is 48 mm/year, and it shows horizontal deformation characteristics with east–west opposing movement centered on the uplift center. The eastern flank of the uplift center is shifting towards the east at a velocity of approximately 15 mm/year, while the western flank of the uplift center is moving in the westward direction at a rate of about 6 mm/year. Moreover, the ground deformation in Area C is near China National Highway 315, potentially posing risks to the road and its surrounding infrastructure. Therefore, it is necessary to enhance the supervision and upkeep of this area. The YSS oilfield is located at the high point of the Youshagou in the northwest of the Youshashan structure in the Qaidam Basin. It is a typical abnormally low-pressure reservoir, so relying solely on natural energy for its development makes it difficult to achieve satisfactory results. Therefore, water injection development has been conducted

in the YSS oilfield since 1996, and it has now transitioned into the late production stage. The long history of water injection and the large amount of water injection required for production maintenance coincide with the serious ground deformation in Areas B and C. However, the GCG oilfield, which initiated trial production in 2020, holds significant oil and gas resources with minimal water injection, resulting in relatively modest ground deformation.

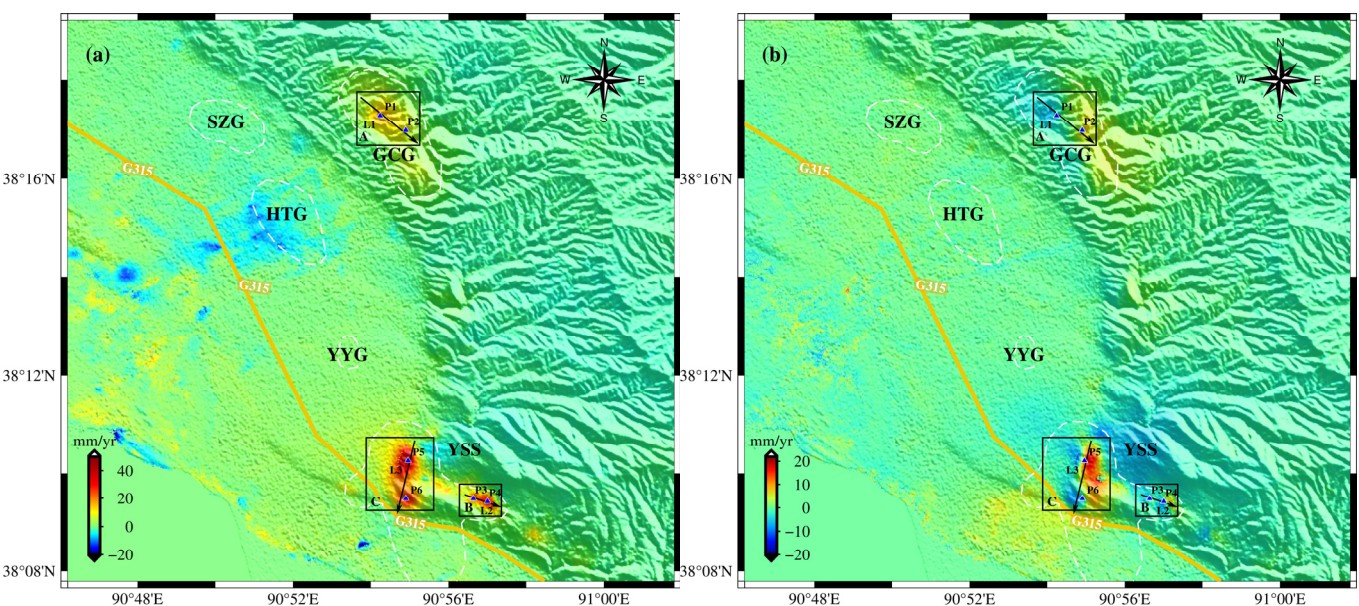

**Figure 5.** (**a**) Vertical deformation rate. Positive and negative values represent uplift and subsidence, respectively. (**b**) East–west deformation rate. Positive values indicate eastward movement, while negative values signify westward motion.

To investigate the spatiotemporal evolution patterns of each uplift in the research area, Figure 6a,b show the time series of cumulative vertical and east–west deformation, respectively, from January 2021 to December 2022. According to Figure 6, Area A was uplifting throughout the observation period, and the uplifted area gradually expanded southward. As of 20 December 2022, the cumulative uplift reached 34 mm. In the horizontal direction, it moved westward significantly from 11 May 2021 to 28 June 2021, and then it resumed its slow movement. Area B exhibited a small relative uplift before September 2021 and a significant increase in relative uplift with time after September 2021, while the east–west deformation always remained relatively stable and slowly increased. Both the vertical and horizontal deformation rates in Area C increased slowly before February 2022 but significantly accelerated thereafter. The maximum vertical displacement was up to 106 mm, and the maximum horizontal displacement was up to 42 mm. Ground deformation is an intuitive reflection of changes in underground reservoir conditions. The greater the intensity of fluid injection into the oil layer, the faster the pore pressure of the reservoir increases. Therefore, we speculate that the variation in the ground deformation rate is related to the variation in injection intensity.

Furthermore, we chose the six points' (P1–P6) (Figure 5) time-series InSAR results to specifically analyze the temporal changes in the oilfields' ground, as illustrated in Figure 7. P1 and P2 are both located in Area A. From their deformation time series, it can be observed that although the surface in Area A has been uplifted compared to January 2021, there is a brief subsidence followed by a rebound every year from May to August. This may be due to a decrease in reservoir pressure caused by oil extraction during this time period, leading to surface subsidence, but quickly rebounding due to the injection of fluid. P3 and P4 are situated on the west and east sides of Area B, respectively. Unlike the roughly linear deformation trend of P3, P4 remained relatively stable before August 2021 but experienced

rapid uplift after August 2021, accompanied by a minor horizontal displacement of less than 5 mm. This may be because the intensity of water injection activities has increased in the west side of Area B since about August 2021. P5 and P6 are situated on the north and south sides of Area C, respectively. They both exhibit a gradual deformation trend before February 2022 and an accelerated deformation trend after February 2022, which is consistent with the increasing injection efforts required for the late-stage production of the oilfield.

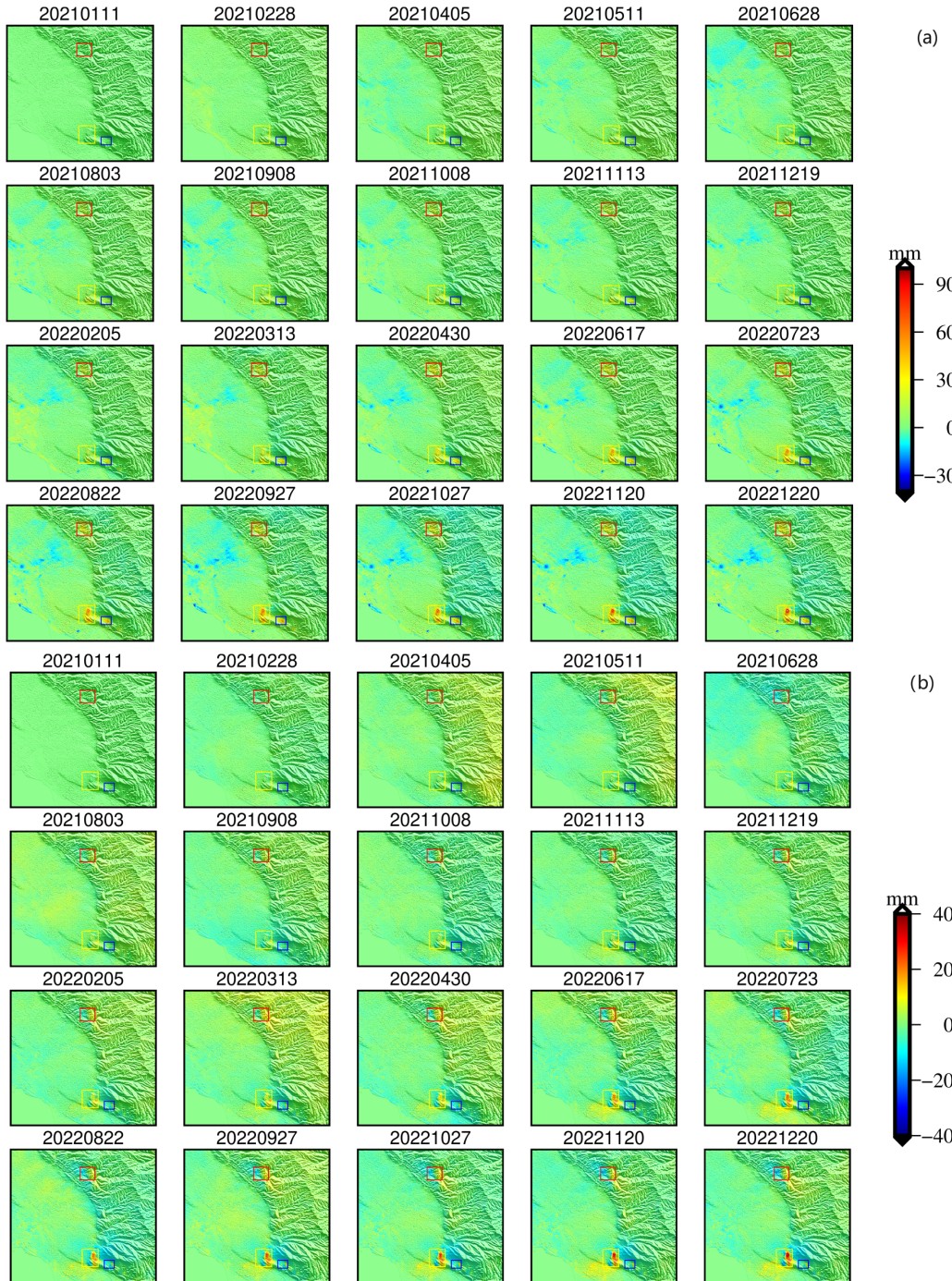

**Figure 6.** Time series of (**a**) vertical and (**b**) east–west cumulative deformation from January 2021 to December 2022 in the study area. The red, blue, and yellow boxes represent Area A, Area B, and Area C, respectively.

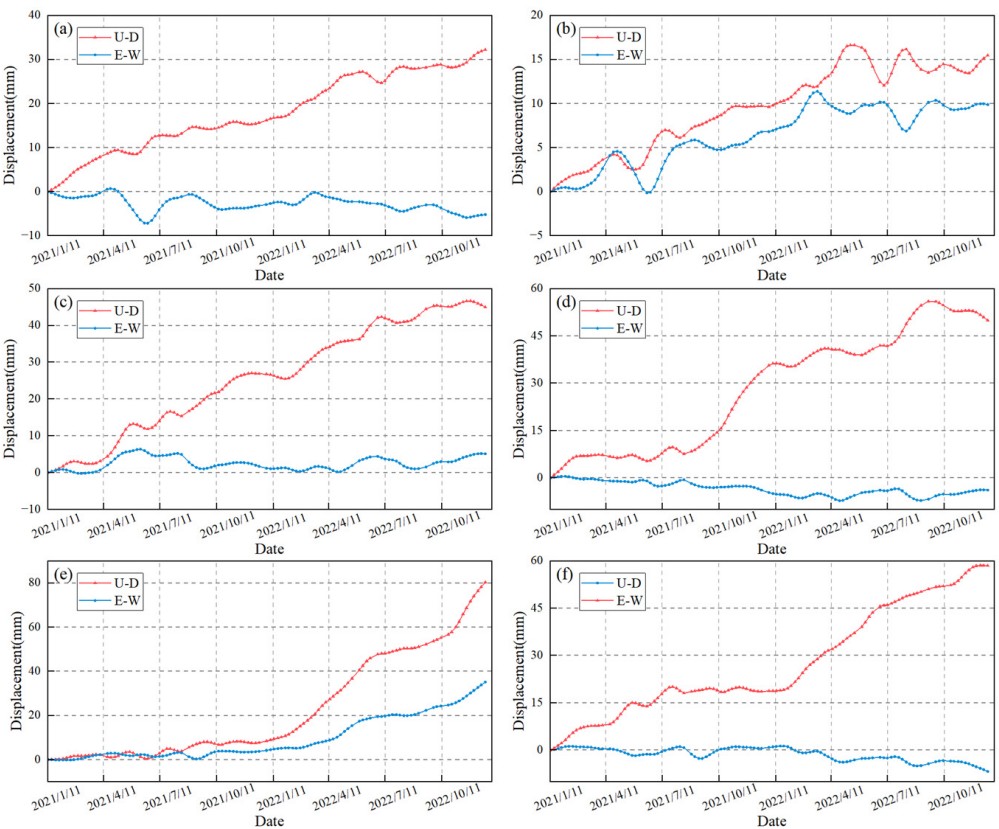

**Figure 7.** The time series of the vertical and east–west deformation at points P1–P6 (**a**–**f**), respectively.

We also plotted three profile lines, L1, L2, and L3, in Areas A, B, and C, respectively. The two-dimensional cumulative deformation along these profile lines is shown in Figure 8. The vertical axis represents the cumulative displacement, and the horizontal axis represents the distance along the direction indicated by the arrows in Figure 5. L1 exhibits an uplift in the vertical direction and reaches its maximum deformation at approximately 1.3 km, after which it gradually decreases. In the horizontal direction, there is a cumulative displacement towards the west before 1.4 km, while there is a cumulative displacement towards the east after 1.4 km. From the analysis above, it can be inferred that there may be a deformation center in Area A in the vicinity of 1.3 km to 1.4 km. Unlike the single peak feature of L1, L2 and L3 exhibit two uplift peaks. Combined with their horizontal deformation characteristics, we speculate that the deformation in Areas B and C is relatively complex and that there may be more than one deformation center.

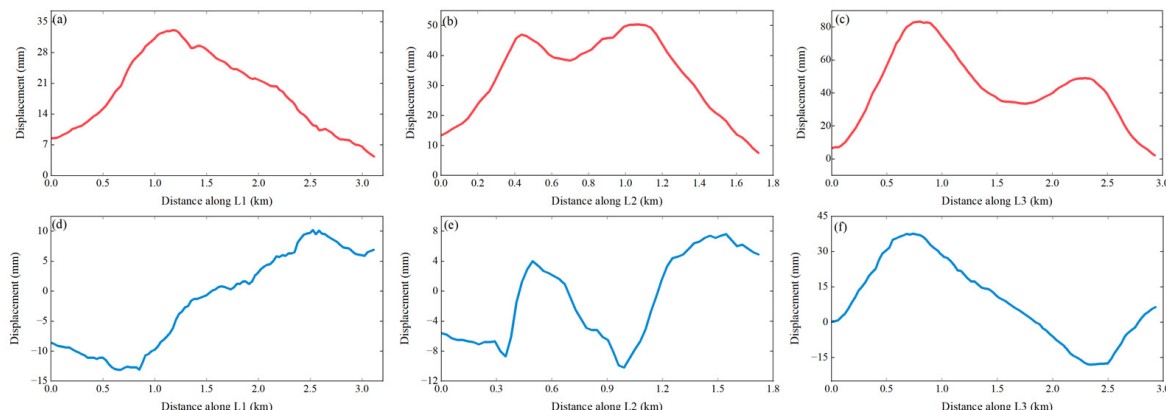

**Figure 8.** The (**a**–**c**) vertical and (**d**–**f**) east–west displacements along the profile lines L1 to L3, respectively.

### 4.2. Reservoir Modeling

Inverting the reservoir parameters based on the deformation information obtained through InSAR technology, combined with corresponding geophysical models, allowed us to achieve a deeper understanding of the underground process of injecting fluids and the reservoir's dynamic behavior. Currently, commonly used geophysical models include the Mogi model [49], the Okada model [44], and the ellipsoid model [42]. The point-source model approximates the deformation as a circle, which differs significantly from the observed deformation, while the Okada model and finite ellipsoid model use more adjustable parameters to simulate complex deformation, which is closer to the observed deformation. Therefore, we chose the dipping dike with uniform opening model in the Okada model and the finite prolate spheroidal model to invert the reservoir parameters in Area B and Area C in this study. Additionally, considering the complex deformation features exhibited in these two areas, this study not only uses the most common single-source model for inversion but also introduces a dual-source model combining a rectangular dislocation surface and a dipping ellipsoid for inversion.

Due to the ambiguity of one-dimensional LOS deformation, it may be difficult for nonlinear inversion to converge quickly. Therefore, we combined InSAR vertical and east–west cumulative ground deformation variables to constrain the inversion, which can increase the robustness of nonlinear inversion. Subsequently, the Bayesian inversion algorithm was used to search for the optimal parameters, and predefined search ranges were set for each parameter. For example, the rectangular length was set from 100 to 1000 m, and the width was set from 10 to 500 m. After 1,000,000 iterations, the parameters reached convergence. In all inversion cases, we typically configured Poisson's ratio to a value of 0.25.

#### 4.2.1. Inversion Result of Area B

We assumed that the ground deformation in Area B is the result of a rectangular dislocation source in the shallow reservoir and used the dipping dike with uniform opening model to invert the oilfield reservoir parameters. The optimal parameters acquired through inversion are shown in Table 2, which lists the inversion results of the rectangular dislocation surface source under Area B. It can be observed that the length and width of the rectangular dislocation surface source are 980.668 m and 399.642 m, respectively. The depth range of the reservoir is 171–227 m, trending from northwest to southeast with an inclination angle of approximately 3°, indicating that it is nearly horizontal. The positive value of the opening suggests reservoir expansion. Figure 9c,d shows the residuals between the observed and modeled deformation in the vertical and horizontal directions. Figure 10 gives the 3D schematic of a rectangular dislocation surface source with optimal parameters in Area B.

**Table 2.** Inversion result of the rectangular dislocation surface source under area B.

| Source | Source Parameter | Optimal Value | Confidence Interval (2.5%) | Confidence Interval (97.5%) |
|---|---|---|---|---|
| | Length (m) | 980.668 | 956.542 | 1016.85 |
| | Width (m) | 399.642 | 372.435 | 433.755 |
| | X | 95.1182 | 78.7143 | 117.508 |
| Rectangular | Y | 255.5 | 232.037 | 279.799 |
| dislocation surface | Depth (m) | 191.863 | 171.431 | 227.192 |
| | Strike (°) | −77.5828 | −79.791 | −74.8579 |
| | Dip (°) | 3.20863 | −0.540244 | −8.16204 |
| | Opening | 0.04148 | 0.03741 | 0.04697 |

We assumed that the ground deformation in Area B arises from the combined effect of a dipping ellipsoid source and a rectangular dislocation surface source in the shallow reservoir. Accordingly, we used the finite prolate spheroidal model and the dipping dike with uniform opening model to invert the reservoir parameters of the oilfield. The optimal

parameters acquired through this inversion are shown in Table 3. For the ellipsoid source, its burial depth was about 218 m. The length of its major and minor semi-axes indicated its morphology as a flat ellipsoid, and its minor axis (30 m) reflected the thickness of the reservoir. The ellipsoid's strike angle relative to the north direction was 65° (clockwise direction), while the dip angle was 17°. DP/mu stands for the ratio of the pressure change to the shear modulus. For the rectangular surface source, its burial depth was about 232 m and its length and width were 728 m and 297 m, respectively. The strike angle was 39° (counterclockwise) and the dip angle was nearly horizontal. The opening of the dislocation surface was 0.063 m. Figure 11c,d shows the residuals between the observed and modeled deformation in the vertical and horizontal directions. Figure 12 presents the 3D schematic of the two sources in Area B.

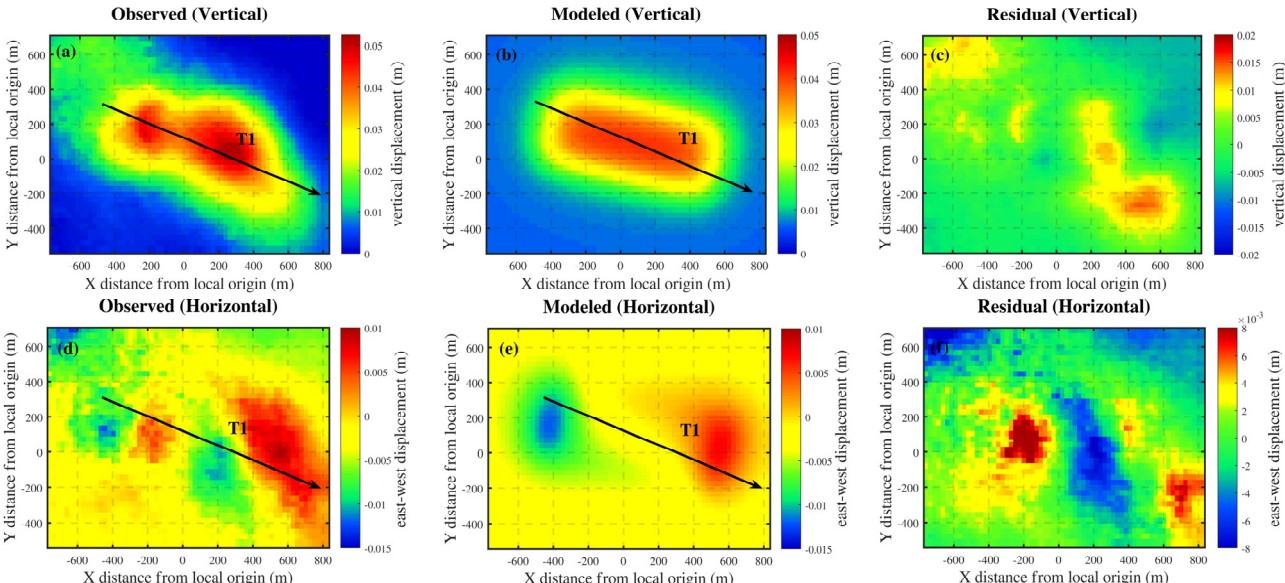

**Figure 9.** The observed deformation field in the (**a**) vertical and (**d**) horizontal directions. The modeled vertical deformation field in the (**b**) vertical and (**e**) horizontal directions; the residuals in the (**c**) vertical and (**f**) horizontal directions.

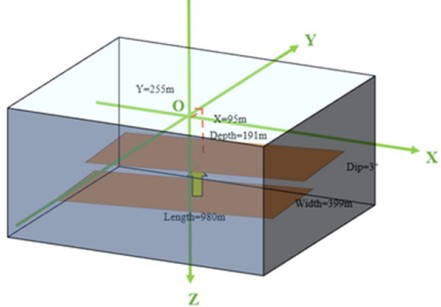

**Figure 10.** (Three-dimensional) 3D schematic of the single deformation source in Area B.

### 4.2.2. Comparative Analysis of Inversion Results for Area B

The residuals between the observed and simulated deformation fields serve as a reliable indicator of the inversion results' quality. From the residual plots in Figures 9 and 11, it can be observed that the residuals in the east–west direction for the single-source model were large and widely distributed, whereas the residuals for the dual-source model were smaller in both directions and mostly existed in the far-field region, which may be due to the influence of the random noise and topography. Additionally, the models used in this study are based on the assumption that they are modeled in a homogeneous elastic half-space, whereas the geology in Area B is somewhat inhomogeneous. Considering the

inherent limitations of the models themselves (e.g., lack of complete rotational freedom), the presence of some residuals, even in the near-field region, is expected.

**Table 3.** Inversion result of two sources under Area B.

| Source | Source Parameter | Optimal Value | Confidence Interval (2.5%) | Confidence Interval (97.5%) |
|---|---|---|---|---|
| Dipping ellipsoid | X | −265.211 | −288.02 | −246.469 |
| | Y | 104.859 | 87.0402 | 126.723 |
| | Depth (m) | 218.904 | 194.592 | 246.927 |
| | Major semi-axis (m) | 206.595 | 168.143 | 252.8 |
| | Minor semi-axis (m) | 15.0091 | 6.77208 | 55.16096 |
| | Strike (°) | 65.0119 | 46.6214 | 75.1686 |
| | Dip (°) | 17.5249 | 10.5561 | 27.3211 |
| | DP/mu | 0.03467 | 0.00378 | 0.09103 |
| Rectangular dislocation surface | Length (m) | 728.471 | 637.649 | 777.356 |
| | Width (m) | 297.448 | 204.269 | 333.208 |
| | X | 459.077 | 413.4 | 476.708 |
| | Y | 50.9312 | 17.552 | 80.1817 |
| | Depth (m) | 232.529 | 211.876 | 286.603 |
| | Strike (°) | −39.4342 | −43.6891 | −34.872 |
| | Dip (°) | −5.77362 | −11.5944 | −0.41442 |
| | Opening (m) | 0.06292 | 0.05257 | 0.09577 |

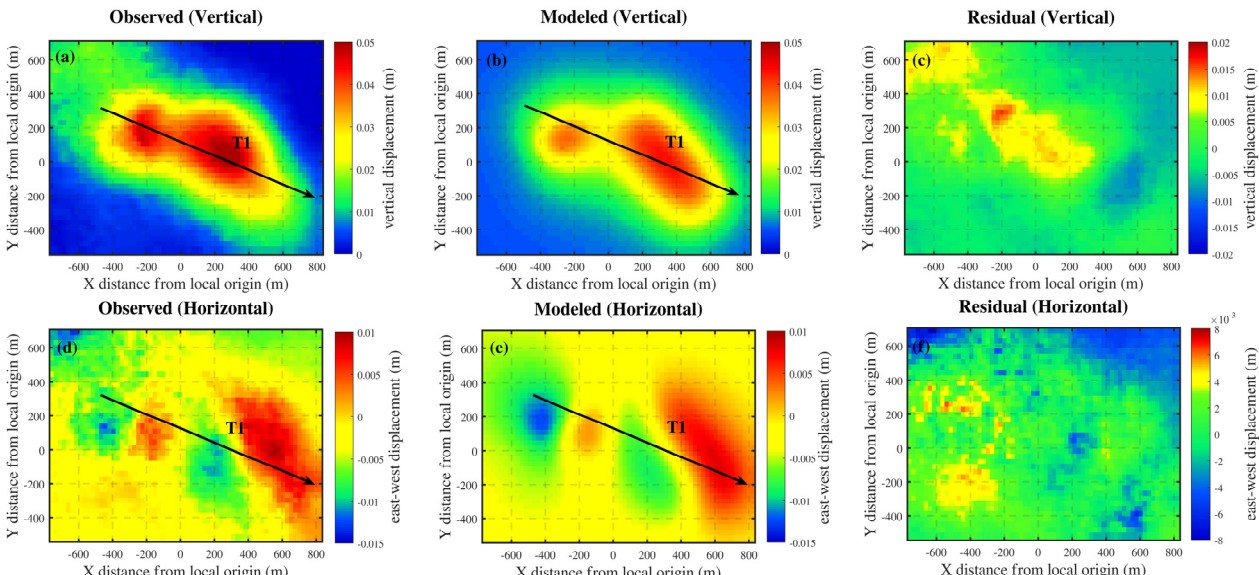

**Figure 11.** The observed deformation field in the (**a**) vertical and (**d**) horizontal directions. The modeled vertical deformation field in the (**b**) vertical and (**e**) horizontal directions; the residuals in the (**c**) vertical and (**f**) horizontal directions.

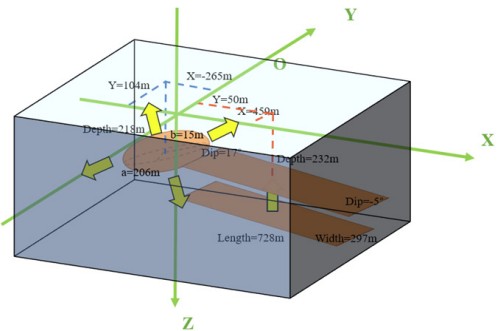

**Figure 12.** Three-dimensional (3D) schematic of the two deformation sources in Area B.

Furthermore, to compare and evaluate the inversion effects of the two models more intuitively and quantitatively, we drew the residual histograms and the deformation profiles along T1 (Figure 11) between their two-dimensional simulated deformation fields and observed deformation fields, as shown in Figure 13. From the distribution of the residuals, it can be seen that the mean and root-mean-square values of the vertical and east–west residuals of the dual-source model were smaller than those of the single-source model. However, compared with the vertical direction, the east–west residuals of both models were smaller and more concentrated towards the zero. This may be due to the fact that the ground uplift caused by the increase in pore pressure is mainly vertical uplift, and the horizontal deformation is smaller and simpler, so the simulation effect of the east–west direction is better than that of the vertical direction. On the other hand, in terms of the fit between the observed and simulated deformations along T1, the simulated curves of the dual-source model are more consistent with the trend of the observed curves, so its degree of fit is higher. In summary, whether from the residual analysis or the fitting degree of deformation, the dual-source model is more suitable for the inversion of reservoir parameters for complex deformations and yields more reliable inversion results than the single-source model.

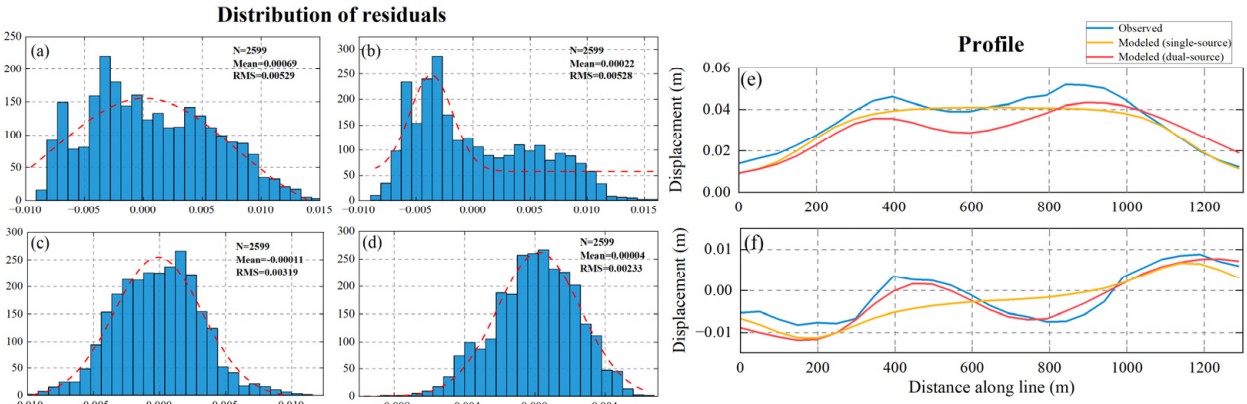

**Figure 13.** (**a**) The distribution of vertical and horizontal residuals for the single-source and dual-source models. (**b**) The distribution of vertical residuals for the dual-source model. (**c**) The distribution of horizontal residuals for the single-source model. (**d**) The distribution of horizontal residuals for the dual-source model. (**e**) The vertical deformation profile along T1. (**f**) The horizontal deformation profile along T1.

4.2.3. Inversion Results for Area C

The comparative analysis of the inversion results of the single-source model and the dual-source model in Area B reveals that the dual-source model is more suitable for inverting the reservoir parameters of complex deformation. Considering the extremely complex deformation in Area C, which cannot be reliably inverted using the single-source model, we directly applied the dual-source model to invert the complex deformation in Area C. The optimal parameters acquired through the inversion are shown in Table 4. It can be seen from the inversion results that the ground deformation in Area C is caused by a rectangular dislocation surface source on the north side of the shallow reservoir and an inclined ellipsoid source on the south side. For the rectangular fault surface source, its length and width are approximately 667 m and 74 m, respectively, with a burial depth of around 451 m. As for the dipping ellipsoid source, its major semi-axis is approximately 728 m, the minor semi-axis is around 17 m, and the burial depth is 546 m. Figure 14c,d shows the residuals between the observed and modeled deformation in the vertical and horizontal directions. A 3D schematic of two sources with optimal parameters for Area C is given in Figure 15. In Figure 16, the root-mean-square values of the vertical and east–west residuals are 6.45 mm and 5.95 mm, respectively, and the fit between the simulated deformations and the observed deformations along the profile T2 (Figure 14) is

also high, which indicates that this inversion is effective and the reliability of the obtained parameters is high, further validating the feasibility of the dual-source model.

**Table 4.** Inversion result of two sources under Area C.

| Source | Source Parameter | Optimal Value | Confidence Interval (2.5%) | Confidence Interval (97.5%) |
|---|---|---|---|---|
| | Length (m) | 667.166 | 39.456 | 703.224 |
| | Width (m) | 74.3788 | 35.9698 | 87.6175 |
| | X | 69.8736 | 44.1824 | 84.5374 |
| Rectangular | Y | 182.243 | 170.078 | 194.093 |
| dislocation surface | Depth (m) | 451.511 | 441.582 | 468.684 |
| | Strike (°) | 15.2082 | 13.0072 | 17.5933 |
| | Dip (°) | 16.0407 | 11.3015 | 19.5505 |
| | Opening (m) | 1.15447 | 1.01517 | 2.39906 |
| | X | 61.8081 | 35.3234 | 151.877 |
| | Y | −1156.77 | −1341.57 | −1113.89 |
| | Depth (m) | 546.16 | 492.207 | 598.769 |
| Dipping ellipsoid | Major semi-axis (m) | 728.4 | 334.435 | 792.369 |
| | Minor semi-axis (m) | 17.2557 | 3.82994 | 55.9840 |
| | Strike (°) | 151.538 | 82.9156 | 158.568 |
| | Dip (°) | −25.2426 | −29.3177 | −9.12962 |
| | DP/mu | 0.07786 | 0.01098 | 0.358436 |

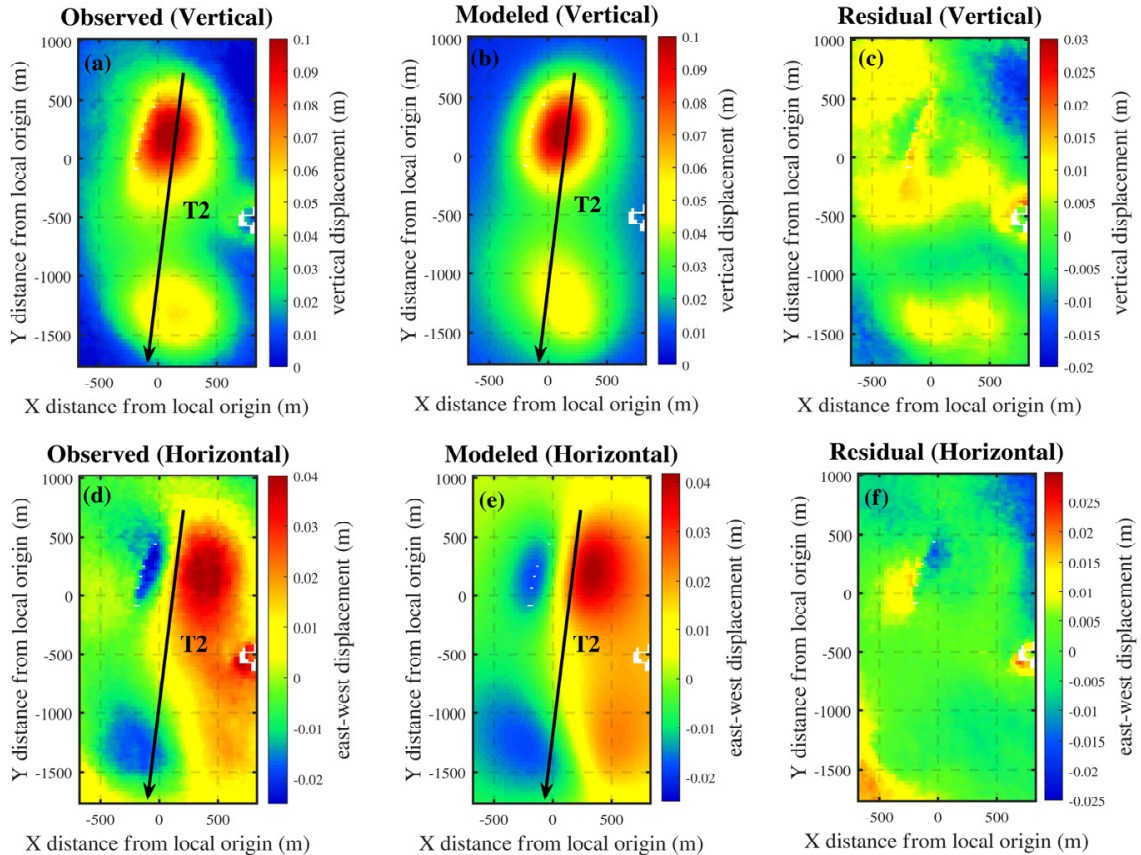

**Figure 14.** The observed deformation field in the (**a**) vertical and (**d**) horizontal directions. The modeled vertical deformation field in the (**b**) vertical and (**e**) horizontal directions; the residuals in the (**c**) vertical and (**f**) horizontal directions.

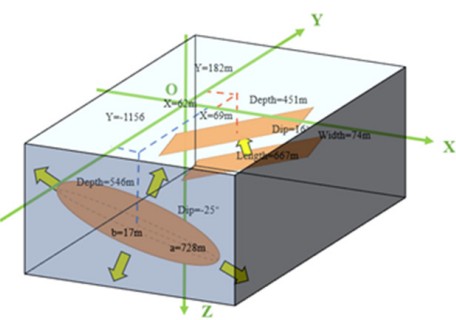

**Figure 15.** Three-dimensional (3D) schematic of the two deformation sources in Area C.

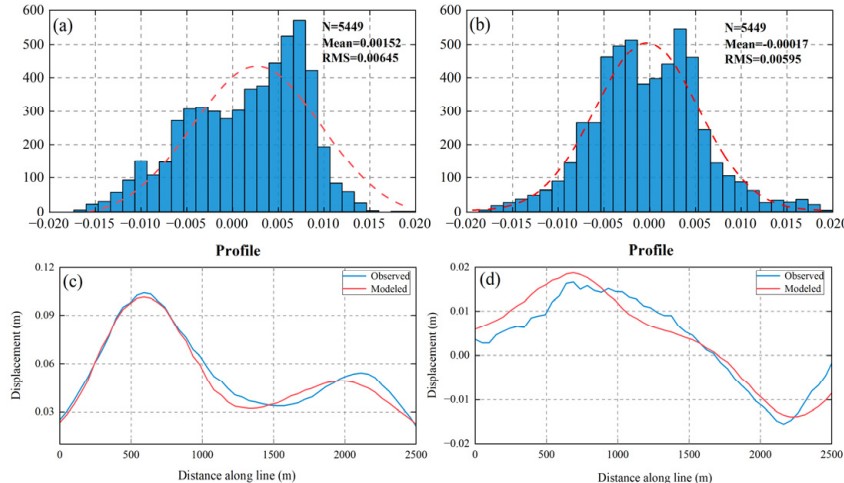

**Figure 16.** (**a**) The distribution of vertical residuals for the dual-source model. (**b**) The distribution of horizontal residuals for the dual-source model. (**c**) The vertical deformation profile along T2. (**d**) The horizontal deformation profile along T2.

## 5. Discussion

In this study, we used the MSBAS method to monitor the ground deformation of the oilfields in the western Qaidam Basin based on Sentinel-1A satellite data. Derived from the SBAS technology, MSBAS technology inherits the technical advantages of SBAS, which can overcome the influence of interferometric phase decorrelation and correct multiple errors, including terrain residuals, orbital errors, and atmospheric delay errors [23]. In addition, regularization or time filtering can remove high-frequency noise and improve the temporal resolution.

Rather than only obtaining the one-dimensional deformation field in the line-of-sight direction, we acquired the two-dimensional time-series deformation fields in the vertical and east–west directions using MSBAS technology. A series of results indicated that in addition to vertical deformation, there was also a significant amount of ground deformation in the horizontal east–west direction in the oilfields, with a maximum east–west deformation of 16 mm/year. Unlike the LOS deformation, which is a mixture of vertical and horizontal deformation, independent vertical and east–west deformations can more comprehensively and intuitively reflect the real deformation effects of the oilfields. This provides more reliable support and guidance for oilfield development planning and land subsidence management.

Subsequently, based on the acquired two-dimensional cumulative deformation, we performed reservoir modeling for two complex deformation areas using geophysical models. In real work scenarios, the injection and production activities in oilfields are often carried out simultaneously at multiple places, resulting in complex ground deformation. By comparing and analyzing the simulation effects of the single-source model and the dual-source model, this study found that there are often multiple deformation sources



underground in areas with complex deformation. The deformation field simulated by the single-source model is relatively simple and cannot effectively reflect the observed deformation field. The dual-source model not only simulates smaller residual mean values and root-mean-square values than the single-source model but also has a better fit with the observed deformation field. Therefore, for areas with complex deformation, the reservoir parameters inverted using the dual-source model are more reliable and better represent the mapping relationship between reservoir changes and ground deformation in oilfields. Moreover, the two-dimensional deformation variable used as the model input variable in this study adds nonlinear inversion constraints that are not found in most current one-dimensional inversion methods, so the inversion efficiency and accuracy are further improved. Unfortunately, due to the lack of water injection and oil production data for the oilfields in the study area, further validation of the inversion results cannot be conducted. In future research on reservoir parameter inversion, efforts should be made to strengthen the collection of such data and the validation of the models.

## 6. Conclusions

This study conducted vertical and east–west deformation monitoring of the oilfields in the western Qaidam Basin by simultaneously processing ascending and descending Sentinel-1A SAR images from 2021 to 2022. Based on this, the reservoir parameters of two deformation areas were inverted using single-source and dual-source models. The main results are as follows:

- In the study area, the HTG oilfield exhibited uneven ground subsidence, with a maximum subsidence rate of 12 mm/year. At the same time, the GCG oilfield and YSS oilfield experienced substantial ground uplift, with a maximum rate of 48 mm/year. Along with this uplift process, the east–west deformation rate in this area also reached 16 mm/year. The cause of the ground uplift in the oilfield may be related to increased reservoir pore pressure resulting from water injection. Changes in water injection intensity may lead to changes in deformation rates.
- Combining the analysis of the inversion results, it can be concluded that the introduction of a two-dimensional deformation field helps improve the non-uniqueness of the inversion results, enhance the robustness of the inversion process and, consequently, obtain more reliable oil reservoir parameters. Furthermore, the dual-source model is more suitable for inverting reservoir parameters of complex deformation compared to the single-source model.

**Author Contributions:** Conceptualization, A.L. and R.Z. (Rui Zhang); methodology, A.L.; investigation, Y.Y. and A.S.; data curation, A.L., T.W. (Tianyu Wang), and T.W. (Ting Wang); writing—original draft preparation, A.L.; writing—review and editing, A.L., R.Z. (Rui Zhang), and X.B.; visualization, R.Z. (Runqing Zhan); funding acquisition, R.Z. (Rui Zhang). All authors have read and agreed to the published version of the manuscript.

**Funding:** This research was jointly funded by the National Key Research and Development Program of China (Grant No. 2023YFB2604001) and the National Natural Science Foundation of China (Grant Nos. 42371460, U22A20565, and 42171355).

**Data Availability Statement:** The Sentinel-1A SAR images can be downloaded from the Alaska Satellite Facility (https://search.asf.alaska.edu/, accessed on 18 November 2023), and the precise orbit data (POD) can be obtained from the European Space Agency (http://aux.sentinel1.eo.esa.int/, accessed on 18 November 2023). The digital elevation model (DEM) is from the United States Geological Survey (https://earthexplorer.usgs.gov/, accessed on 18 November 2023). The open-source software MSBAS is available at https://insar.ca/multidimensional-small-baseline-subset-msbas/ (accessed on 18 November 2023). The GBIS software package used for parameter inversion is from the Centre for Observation and Modelling of Earthquakes, Volcanoes, and Tectonics (COMET) (https://comet. nerc.ac.uk/gbis/, accessed on 18 November 2023). We are very grateful for the above support.

**Acknowledgments:** We appreciate the European Space Agency for providing the Sentinel-1A data freely. We are also grateful to the editor and reviewers for giving us valuable suggestions and comments to help us improve this paper.

**Conflicts of Interest:** The authors declare no conflicts of interest.

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
