# Peer review of "Oilfield Reservoir Parameter Inversion Based on 2D Ground Deformation Measurements Acquired by a Time-Series MSBAS-InSAR Method"

_remotesensing, doi:10.3390/rs16010154_

Round 1

Reviewer 1 Report

Comments and Suggestions for Authors

In this study, the MSBAS TS-InSAR method was applied to reveal the deformation and inverse the parameter of oilfield reservoir in Qaidam Basin based on Sentinel-1A SAR images during 2021-2022. The 2D deformation which derived from Sentinel-1A SAR images was used to improving the robustness of oilfield reservoir parameter inversion. The results first reveal the deformation trend by InSAR measurements over Qaidam Basin and provided the reference method for reservoir parameter inversion in other oilfields. Here three suggestions for this manuscript.

1.     The resolution of Sentinel-1 SAR dataset is not centimeter, please revise it in Table1;

2.     The GCG,HTG,and YSS should be marked in Fig5;

3.     The author should add some details about the accuracy of InSAR measurement, such as the comparing the vertical deformation for ascending track and descending tracks in section 4.

Reviewer 2 Report

Comments and Suggestions for Authors

This paper utilizes the MSBAS method to extract two-dimensional deformations over the oilfields in the western Qaidam Basin. Subsequently, reservoir parameters for the complex deformations in the oilfield are inverted using a combination of multi-source and single-source models. The research presented in this paper effectively broadens the application of InSAR technology in the monitoring of oilfields and reservoir parameter inversion. Two-dimensional deformation monitoring in time series may more accurately depict the actual deformation characteristics of the oilfield. The comparison of the inversion results between multi-source and single-source models holds significant reference value for model selection in similar studies. The overall structure of the paper is clear and presents the entire experimental process in a coherent manner.

I think the outstanding performance of this paper is commendable. Below are some comments:

1) It is recommended to include more literature that introduces the background of the western Qaidam Oilfield, emphasizing the significant oil production in this region and the necessity of monitoring oilfield deformations.

2) Please label areas A, B, and C in Figure 6 to provide a more intuitive understanding of the analysis for the corresponding areas.

3) Please refine the language of the paper to enhance its academic quality.

4) In the literature review, ensure to cite the latest research to maintain the paper's cutting-edge nature.

5) If possible, please furtherly analyze the time-series deformation characteristics in conjunction with real injection-production data of the study area and validate the inverted parameters to further enhance the credibility of the paper.

Reviewer 3 Report

Comments and Suggestions for Authors

The article is written in an appropriate way. It is clear. The results are weel described.

Please take care of mathematical symbilism because some symbols are used to identify different quantitities.

I suggest to add some citation about DInSAR literature:

Please on page 2 Line 57-58 when referring to SBAS, PS and DSInSAR add the references:

1) for PS : A. Ferretti, C. Prati and F. Rocca, "Permanent scatterers in SAR interferometry", IEEE Trans. Geosci. Remote Sens., vol. 39, no. 1, pp. 8-20, Jan. 2001.

2) for SBAS : P. Berardino, G. Fornaro, R. Lanari and E. Sansosti, "A new algorithm for surface deformation monitoring based on small baseline differential SAR interferograms", IEEE Trans. Geosci. Remote Sens., vol. 40, no. 11, pp. 2375-2383, Nov. 2002.

3) for DS-InSAR : A. Ferretti, A. Fumagalli, F. Novali, C. Prati, F. Rocca and A. Rucci, "A new algorithm for processing interferometric data-stacks: SqueeSAR", IEEE Trans. Geosci. Remote Sens., vol. 49, no. 9, pp. 3460-3470, Sep. 2011.

4) for DS-InSAR : G. Fornaro, S. Verde, D. Reale and A. Pauciullo, "CAESAR: An approach based on covariance matrix decomposition to improve multibaseline-multitemporal interferometric SAR processing", IEEE Trans. Geosci. Remote Sens., vol. 53, no. 4, pp. 2050-2065, Apr. 2015.

5) for DS-InSAR :G. Fornaro, A. Pauciullo, D. Reale and S. Verde, "SAR coherence tomography: A new approach for coherent analysis of urban areas," 2013 IEEE International Geoscience and Remote Sensing Symposium - IGARSS, Melbourne, VIC, Australia, 2013, pp. 73-76, doi: 10.1109/IGARSS.2013.6721095.

In paragraph 3.1, when talking about LOS decomposition in vertical, east-west, and components please consider citing also :

Cascini L, Fornaro G, Peduto D (2010) Advanced low- and full-resolution DInSAR map generation for slow-moving landslide analysis at different scales. Eng Geol 112:29–42.

Finally in the introduction take into consideration citing , whne talking about SBAS processing of Sentinel-1 data and the potentialities related to atmospheric component estimation.

N. Pierdicca et al., "Excess Path Delays From Sentinel Interferometry to Improve Weather Forecasts," in IEEE Journal of Selected Topics in Applied Earth Observations and Remote Sensing, vol. 13, pp. 3213-3228, 2020, doi: 10.1109/JSTARS.2020.2988724.
